# Loneliness and Isolation in the Era of Telework: A Comprehensive Review of Challenges for Organizational Success

**DOI:** 10.3390/healthcare13161943

**Published:** 2025-08-08

**Authors:** Elisabeth Figueiredo, Clara Margaça, José Carlos Sánchez-García

**Affiliations:** 1Faculty of Psychology, University of Salamanca, 37005 Salamanca, Spain; efigueiredo@usal.es (E.F.); jsanchez@usal.es (J.C.S.-G.); 2Department of Psychology, University of Valladolid, 47002 Valladolid, Spain

**Keywords:** telework, loneliness, isolation, productivity, innovation, knowledge sharing, organizational performance, bibliometric review

## Abstract

**Purpose:** As remote work becomes increasingly prevalent, scholarly and organizational attention has largely centered on stress and workload. However, emerging evidence highlights loneliness and isolation as critical yet underexplored factors with profound implications for employees’ mental health and organizational performance. **Method:** This study conducts a bibliometric review of 65 peer-reviewed articles indexed in the Scopus database, following PRISMA guidelines and employing VOSviewer for data analysis and visualization. The objective is to examine how loneliness and isolation in remote work settings affect psychological well-being and productivity. **Findings:** The findings reveal that the absence of physical interaction and structured social environments exacerbates feelings of detachment, contributing to emotional strain and reduced job performance. Despite the growing importance of remote work, the literature remains fragmented in addressing its broader psychological and organizational consequences. **Originality/Value:** This study offers theoretical insights and practical recommendations for managers and policymakers, emphasizing the need for preventive strategies and inclusive management practices to support employee well-being, foster engagement, and sustain organizational effectiveness in remote work contexts.

## 1. Introduction

In recent years, telework has become increasingly prevalent across organizations worldwide [1], driven by globalization, advances in communication technologies [2], and the COVID-19 pandemic [3]. While telework is often praised for benefits such as flexibility, autonomy, and reduced operational and commuting costs [4,5], it also has introduced significant shifts in work dynamics [6,7,8], raising concerns about its impact on employees’ mental health [9,10,11]. Despite its undeniable advantages, remote work, by altering routines and reducing social interaction, can generate significant consequences for workers’ mental health, including mood swings [6], irritability, anxiety, and difficulty concentrating, aggravated by unclear boundaries between personal and professional life. Furthermore, sleep disturbances have also been reported, potentially due to the lack of physical separation between workspaces and rest areas, as well as increased screen time and reduced exposure to natural light. Although flexibility is part of this work model, declining motivation and psychological fatigue have triggered productivity challenges [12], as interpersonal interaction and team cohesion are essential for maintaining emotional resilience and job satisfaction. Another challenge encountered is the lack of prior experience with teleworking, revealing greater difficulties in adapting to the work-from-home model, which, in turn, negatively impacts psychological well-being [7,10]. Directly related to these issues is the lack of adequate ergonomic and organizational support systems. From an organizational perspective, Ref. [8] observed that the sudden transition to remote work required significant changes in HR practices, and not all organizations were able to implement effective support systems. This body of evidence highlights the need for structured telework policies, adequate support systems, and proactive mental health strategies to mitigate the negative impacts of remote work.

The literature points to increased psychosocial risks associated with telework, particularly stemming from reduced workplace connection, feelings of loneliness [7], and both physical and social isolation [10,12,13]. When prolonged, these conditions can evolve into serious psychological disorders such as burnout (exhaustion due to overinvestment) [14] or boreout (exhaustion due to boredom) [15,16].

Prolonged physical distance from the workplace remains a risk factor for well-being, even with virtual connectivity. One prominent consequence is the decline in spontaneous social interactions typical of in-person work environments [17,18]. Refs. [19,20] details how reduced interpersonal contact and the perceived lack of social and emotional support exacerbate feelings of isolation, anxiety, and disconnection and diminish personal fulfillment [18,21]. Informal social interactions, previously fostered by hallway conversations, coffee breaks, or shared meals, play a crucial role in emotional regulation and psychological well-being [22].

The psychological consequences of loneliness and isolation have gained increasing attention in discussions on emerging work models [23]. Studies consistently link loneliness with higher risks of depression, anxiety, stress, and burnout [14,24,25]. Social disconnection contributes to negative thought patterns and impaired emotional [22], while chronic loneliness is associated with cognitive deficits that hinder concentration and memory, factors that may compromise professional performance [26,27]. Additionally, blurred boundaries between personal and professional life in remote settings often lead to emotional overload and further increase burnout risk [27]. Without appropriate interventions, these effects may have long-term repercussions for individual and organizational well-being [23,28].

Although teleworking has been widely studied, its impact on workers’ mental health and organizational outcomes has only recently received scholarly attention. A significant gap remains in understanding how loneliness and isolation among teleworkers affect psychological health and, in turn, organizational performance and sustainability [29]. As research in this field diversifies, there is a growing need to synthesize available evidence for a more comprehensive understanding of these dynamics. To address this, a systematic review of 65 articles was conducted, aiming to analyze the relationship between loneliness, isolation, and psychological disorders, and to investigate their potential impacts on the development and success of organizations employing remote workers. The Scopus database was used for data collection, guided by the PRISMA methodology. VOSviewer [30] software facilitated data visualization and mapping. Following [31], the study posed the following research questions:

RQ1: How have scientific publications on this topic evolved over time?

RQ2: What effects do loneliness and isolation have on the mental health of teleworkers, and how does the literature reflect these impacts?

RQ3: What are the predominant theoretical approaches in the literature?

RQ4: What organizational strategies can mitigate the negative effects of isolation and loneliness among remote workers?

This study aims to offer novel insights for organizations and managers, emphasizing the importance of developing effective internal policies to reduce the adverse effects of isolation and loneliness in teleworking and to promote enhanced employee performance and organizational resilience.

## 2. Methodology

This study adopts a bibliometric review approach to examine the scientific literature on the psychological and organizational impacts of loneliness and isolation in remote work contexts. A total of 65 peer-reviewed articles indexed in Scopus database were analyzed, given Scopus’s recognized comprehensiveness and indexing quality across diverse fields of knowledge [32]. To identify thematic trends and research patterns, bibliometric mapping was conducted using VOSviewer software v.1.6.17 [30,33], which enables the visualization of co-citation networks and keyword co-occurrence, facilitating the identification of research clusters and emerging topics. The review process adhered to the PRISMA (Preferred Reporting Items for Systematic Reviews and Meta-Analyses) guidelines, ensuring methodological transparency and replicability through the sequential steps of identification, screening, and inclusion (Figure 1).

### 2.1. Search Strategy

The search strategy employed a combination of keywords relevant to remote work, social isolation, and organizational performance. The following Boolean search string was used: TITLE-ABS-KEY (telework* OR “home office” OR “work from home” OR telecommuting OR “virtual work” OR “remote work”) AND TITLE-ABS-KEY (“loneliness” OR isolation OR solitude OR “social isolation”) AND TITLE-ABS-KEY (performance OR productivity OR “work performance” OR “organizational performance”). Filters were applied to include only articles published in English, Portuguese, French, and Spanish between 2000 and 2024. The four languages were selected based on their relevance and comprehensiveness in academic contexts, and in the countries where telework has been most widely implemented. This choice aims to ensure an inclusive and unbiased synthesis of evidence across diverse contexts. The timeframe from 2000 to 2024 was defined to reflect the period during which digital technologies have increasingly transformed the workplace. The beginning of the 21st century represents a pivotal moment in the expansion of teleworking [34], making this span particularly suitable for capturing both major developments and the most recent trends in the field.

### 2.2. Inclusion Criteria

Peer-reviewed articles that explored the themes of loneliness and/or isolation within the context of teleworking, remote work, or distance work were included in this review. Studies were considered eligible regardless of the geographical context of the study, as long as they were published between 2000 and 2024 and written in one of the four previously defined languages.

To ensure methodological and conceptual rigor, this review applied consistent definitions for the key constructs. Telework was defined as the performance of professional activities outside the employer’s premises, enabled by digital communication technologies [12,35]. Only studies involving participants engaged in continuous telework were included. Loneliness was understood as a negative emotional experience resulting from the perceived discrepancy between desired and actual social relationships [26]. Eligible studies employed validated scales or provided clear, explicit reports of loneliness. Isolation was defined as the absence or reduced frequency of social contact and interaction [36]. Only studies that clearly distinguished isolation from loneliness and addressed the concept in a work-related context were considered.

### 2.3. Exclusion Criteria

Conference abstracts, book chapters, theses, dissertations, and non–peer-reviewed articles were excluded from this review. Studies that focused solely on physical isolation, such as technical or infrastructural barriers, without addressing psychosocial dimensions were not considered. Articles examining populations outside the working-age range (e.g., older adults, children, or students) or exploring isolation in non-work contexts were also excluded. In addition, duplicate records and studies without full-text availability were removed from the final selection.

### 2.4. Data Selection and Extraction

The initial search yielded 170 articles. After removing duplicates and applying the inclusion and exclusion criteria, 97 articles were retained for full-text review. Of these, 65 were deemed directly relevant for final analysis, focusing explicitly on the impact of loneliness and perceived isolation on psychological well-being and organizational outcomes (Table 1). The screening process involved three stages, title review, abstract screening, and full-text assessment, conducted independently by two reviewers to ensure reliability. Disagreements between reviewers were resolved through discussion and consensus. If necessary, a third reviewer was consulted to reach a final decision. However, no formal statistical estimation of inter-rater reliability, such as Cohen’s kappa coefficient, was conducted. Data extraction followed a systematic approach, focusing on variables aligned with the study’s guiding research questions. Bibliometric data, including authorship, year of publication, journal, and number of citations, were collected to analyze the temporal evolution of publications and assess the academic impact of the scientific output (RQ1). The geographic context of each study was also recorded to support the analysis of regional patterns in research activity. Additionally, several bibliometric variables were analyzed, such as co-authorship (based on the number of shared documents), contributing countries, citation patterns, co-citation networks (i.e., instances where two documents are cited by the same source), and the co-occurrence of keywords. These analyses provided a more nuanced understanding of the structure and development of the literature on the topic. We also examined the sociodemographic characteristics of the study samples, such as the professional sector, to explore how these factors relate to the effects of loneliness and isolation on workers’ mental health (RQ2). Furthermore, the methodological design of each article was described, including study type, procedures, and theoretical frameworks, to identify the main conceptual foundations used across the literature (RQ3). Finally, results concerning the psychological impacts of loneliness and isolation, along with the organizational strategies proposed to mitigate these negative effects, were extracted and organized through thematic and bibliometric analysis. This process involved identifying thematic clusters through co-occurrence and network analysis, facilitating the organization of the data into interconnected knowledge domains (RQ1 and RQ4).

## 3. Results

### 3.1. Descriptive Analysis

A temporal analysis of publication trends (Figure 2) illustrates a marked increase in research interest from 2020 onward, with a publication peak in 2022, reflecting the heightened relevance of remote work during the COVID-19 pandemic. Before 2020, scholarly attention to the topic was minimal, with isolated publications in 2000 and 2015. The number of publications rose to 12 and peaked at 17 in 2022, remaining steady through 2024. This trend underscores the post-pandemic urgency to better understand the psychosocial consequences of large-scale telework adoption [10].

Citation analysis further highlights peaks in academic influence during 2000, 2015, and 2021, with notable citation counts in 2015 (approximately 970 citations) and 2021 (approximately 860 citations). These peaks likely correspond to seminal works that spurred subsequent research activity. Between 2001 and 2013, citation activity was negligible. A steady increase began in 2019, reaching a peak in 2021, followed by a decline in citation volume through 2024, potentially reflecting the lag time between publication and academic uptake of newer studies.

The geographical distribution of the studies analyzed indicates a predominance of research originating from Europe and North America. In total, the 65 reviewed studies span 40 countries, with the United States emerging as the most frequently represented nation (n = 9), followed by Germany, the United Kingdom, and India, each contributing six articles (Figure 3).

Table 2 provides an overview of the methodological approaches and sample characteristics. Quantitative methodologies were employed in the majority of studies (73.84%), followed by qualitative approaches (12.31%). The research samples encompass a broad range of professional sectors and work contexts, including academic staff, healthcare professionals, public sector employees, industrial workers, and financial sector employees. Among these, the category of “general workers” appears most frequently in both quantitative and qualitative research.

A review of publication outlets shows that *Frontiers in Psychology*, the *International Journal of Environmental Research and Public Health*, and the *Journal of Occupational and Environmental Medicine* each published three of the reviewed articles, leading in terms of publication volume. Table 3 presents the 25 journals with the highest representation in this review, based on data from the Scimago Journal & Country Rank (SJR) and the Journal Citation Report (JCR). The fields of Business, Management and Accounting, and Social Sciences jointly account for the highest share of publications (17.6% each), followed by Medicine (16.7%) and Psychology (13%) (Figure 4).

Among the journals analyzed, *Psychological Science in the Public Interest* (JIF 18.2), *Mayo Clinic Proceedings* (JIF 7.9), and the *Journal of Occupational Health Psychology* (JIF 5.9) stand out due to their high impact factors. Furthermore, 37 of the selected articles (63%) were published in journals ranked in the first or second quartile (Q1 and Q2), underscoring the growing scientific recognition and relevance of publications belong to quartiles 1 and 2, representing 63% of the total research domain.

The bibliometric analysis also highlights the influence of key publications in shaping the field. Among the 65 reviewed articles, the 5 most cited publications are identified as the most impactful based on total citation counts and citations per year:[37]: 970 citations; 97.0 citations per year.[38]: 514 citations; 128.5 citations per year.[39]: 442 citations; 17.7 citations per year.[36]: 215 citations; 43 citations per year.[40]: 176 citations; 44 citations per year.

Table 4 illustrates the distribution of publications by author between 2000 and 2024. Most authors (n = 150) contributed only one article to the topic. Only [36] authored more than one paper. Their 2020 publication, with 215 total citations and an average of 43 citations per year, ranks fourth among the most cited works. The most cited article per year is by [38], with an average of 128.5 citations per year, reflecting its high influence.

### 3.2. Topic Clusters

To map the thematic structure of the field, a keyword co-occurrence analysis was conducted. A total of 488 words were identified and grouped into 22 clusters. The most frequently occurring terms were COVID-19, teleworking, social isolation, mental health, job satisfaction, and productivity. As shown in Figure 5, these terms are visualized as larger nodes, indicating higher frequency and stronger interconnections.

To focus the analysis and improve interpretability, a threshold of at least three keyword co-occurrences was applied, resulting in 47 keywords organized in 6 distinct thematic clusters (Figure 6), generated by VOSviewer software [33]. In this study, each cluster represents a distinct thematic core within the body of research reviewed, reflecting specific aspects of loneliness and isolation in the context of teleworking. The interpretation of these thematic groupings was informed by a close, critical reading of the terms associated with each cluster, supported by a qualitative examination of the articles in which these terms appeared most prominently and consistently. Through this analytical process, six main thematic axes were identified, namely (1) mental health and work performance; (2) COVID-19 and impact on work; (3) stress and work engagement; (4) social relationships, isolation, and support in remote work; (5) working conditions and professional isolation; and (6) telework, productivity, and sustainability (see Table 5). These clusters are highly interconnected, reflecting the complex, multidisciplinary nature of the topic. The keyword “COVID-19” serves as a central node, linking multiple clusters and highlighting its pivotal role in accelerating telework adoption and shaping related psychosocial and organizational outcomes. “Telecommuting” and “social isolation” emerge as crucial connecting elements between the purple and green clusters, while “mental stress” links the red, blue, and yellow clusters. The term “Productivity” is strongly associated with teleworking, job performance, and social isolation, underscoring its significance in debates around the consequences of remote work.

The distribution of focus across the reviewed literature is as follows: 26.2% of the studies address mental health and work performance, 18.1% examine the impact of COVID-19 on work, 16.4% explore stress and work engagement, 14.3% focus on social relationships, isolation, and support, 13.8% investigate working conditions and occupational isolation, and 11.2% analyze telework in relation to knowledge sharing, productivity, and sustainability.

Table 5 presents a summary of the clusters and associated keywords.

#### 3.2.1. Cluster 1—Mental Health and Job Performance

The first cluster, in red, encompassing 26.2% of the analyzed articles, centers on the relationship between teleworking and mental health. It brings together terms such as anxiety, loneliness, depression, stress, psychological distress, and work performance. This thematic grouping reflects the complexity of the factors influencing psychological well-being and their direct implications for professional performance. Within this cluster, numerous studies emphasize loneliness as a mediating factor linking psychological disorders and reduced productivity [6]. Recent research has therefore focused on understanding the mechanisms underlying loneliness, particularly its effects on psychological well-being and work efficiency [44]. Ref. [23] define loneliness as a subjective experience marked by perceived isolation or disconnection from one’s environment. In this workplace context, this occurs when an individual feels socially distanced or emotionally detached from the organization, even while interacting with others [41]. Ref. [45] distinguishes between emotional loneliness—stemming from a lack of affective bonds with colleagues—and social loneliness—resulting from infrequent workplace interactions.

Loneliness in professional settings often diminishes employees’ commitment and sense of belonging to the organization [22]. In telework scenarios, Ref. [46] highlights the impact of poorly adapted technological tools, noting that when ICTs are not aligned with task requirements, professionals are more likely to experience loneliness and underperformance. Conversely, technologies that demonstrate task–technology fit can mitigate these negative effects.

A review by [47] found that prolonged periods of teleworking exacerbate feelings of loneliness and heighten perceived separation from the organization, as remote workers lack the same opportunities for social engagement as their in-office counterparts. Similarly, Ref. [48] emphasizes the detrimental psychological consequences of teleworking, which can intensify emotional vulnerability and lead to a form of psychological distress characterized by emotional strain, depersonalization, and reduced personal fulfillment.

In a quantitative study with 994 employees in Germany, Ref. [49] observed that the benefits of remote work are not sustained indefinitely. Extended remote work was associated with anxiety, depression, isolation, reduced job satisfaction, and productivity losses stemming from a lack of technical and emotional support [1]. In a large-scale Dutch study, Ref. [50] found that many employees attributed mental health issues to the obligation to work from home and the resulting social isolation. Similarly, Ref. [24], examining Indian teleworkers during the COVID-19 pandemic, found a significant correlation between isolation and heightened anxiety (*p* = 0.005), as well as between anxiety and stress (*p* = 0.000).

Refs. [6,28] argue that one of the main challenges of teleworking lies in the social disconnection it fosters, which often leads to dissatisfaction and demotivation. According to these authors, virtual work models tend to weaken the relational ties between employees and organizations, diminishing engagement and identification with their employer.

#### 3.2.2. Cluster 2—COVID-19 and Impact on Work

The second cluster, in green, focuses on the relationship between remote work, emotional burnout, and the influence of both work and home environments on employee performance. It also explores strategies for prevention and control aimed at mitigating the negative consequences of this work model. The COVID-19 pandemic stands out as a pivotal element in this cluster, serving as a catalyst for the widespread adoption of telework. Its centrality in Figure 6 underscores its significance in current debates, highlighting its broad social, psychological, and organizational repercussions.

Teleworking has profoundly altered professional routines, introducing new challenges [8], among which emotional burnout has gained prominence [16]. The pandemic period saw a notable surge in stress and emotional exhaustion among remote workers [27]. Burnout, which is characterized by a persistent negative emotional state, stems from chronic work-related stress [1], task overload [48], reduced social interaction, and erosion of work–life boundaries [51]. According to [14], burnout becomes particularly acute in teleworking contexts, where mental fatigue is often aggravated by social isolation and limitations in digital communication tools [1,3].

Although traditionally associated with sectors such as healthcare, education, and service professions, burnout is increasingly affecting remote workers due to prolonged isolation [10]. As noted by [52], the telework context often amplifies burnout, heightened productivity expectations, and time management difficulties. Similarly, Refs. [10,11] emphasize the continuous pressure for results as a factor that undermines employee well-being.

Studies by [1,11,53] demonstrate that the coexistence of professional and personal demands in shared spaces intensifies tensions and contributes to family conflicts. These conflicts are more prevalent in in-home office settings, where role overlap directly impacts productivity and psychological well-being [37]. The resulting functional overload compromises work–life balance and deteriorates workers’ quality of life [51,52].

A closely related phenomenon is technostress, which refers to stress induced by intensive use of digital technologies in the workplace [7]. As teleworking expanded, employees were required to manage an increasing number of digital platforms and tools [8,54,55], often leading to cognitive overload [10] and phenomena such as “Zoom fatigue” [56]. The expectation of constant connectivity and availability is a central contributor to technostress [7]. Inadequate digital competencies and the ongoing demand for technological adaptation have further elevated stress and anxiety levels [1]. As noted by [57], the absence of a structured work environment can exacerbate technostress and contribute to increased psychological suffering. These effects have clear implications for performance, reducing productivity and harming mental resilience, especially among professionals struggling with digital tools [1].

Consequently, prevention and control strategies to address these challenges are increasingly emphasized [1]. Organizations must implement clear mental health support policies that safeguard work–life boundaries [58]. Ref. [57] recommends both individual and organizational psychological support to counter burnout and technostress and to promote greater harmony between personal and professional responsibilities. Additionally, continuous training in digital skills [59] is particularly relevant to alleviate technostress and enhance worker confidence. Creating virtual spaces for informal interaction, such as online social events and digital team-building activities, also plays a key role in fostering employee engagement and well-being [39].

#### 3.2.3. Cluster 3—Professional Stress and Work Engagement

The third cluster, in blue, highlights the impact of social isolation as an objective condition characterized by the reduction or absence of regular interpersonal interactions [23]. When linked to remote work, this isolation has been identified as a risk factor for increased stress, as it limits opportunities for social exchange—an essential element for emotional support and the development of a healthy organizational climate. According to [60], physical distancing and the absence of face-to-face interactions among employees are critical elements that reduce work engagement, hinder collaboration, stifle creativity, and disrupt knowledge sharing, core processes for sustained and effective organizational performance.

Social isolation and the transition to remote work have had a direct and observable impact on work engagement [29], a construct defined by [61] as the enthusiastic and energetic involvement of workers in their professional tasks, characterized by vigor, dedication, and absorption. The physical separation from the workplace and the reduction in contact with colleagues weaken motivation and jeopardize the sense of organizational belonging [58], thereby reducing employees’ commitment to task execution and the achievement of organizational goals [22]. Ref. [36] emphasizes that social isolation, especially when compounded by stress, affects psychological well-being and reduces work engagement, potentially triggering a downward spiral of reduced productivity and emotional exhaustion [28].

Occupational stress is among the most prevalent work-related conditions and has profound effects on worker health. It is typically defined as an emotional, physical, and cognitive response to work demands that exceed the individual’s resources [62]. The Job Demands–Resources (JD-R) model, developed by Demerouti et al. (2001) [63], explains that stress emerges when job demands exceed the available resources needed to manage them. This model has been widely used to explore the negative relationship between occupational stress and work engagement, consistently demonstrating that prolonged stress significantly erodes employee engagement levels.

Ref. [62] argues that remote work can reduce occupational stress, but only when it is not perceived as socially isolating. Sources of stress in teleworking contexts include deadline pressure, interpersonal tensions, and a lack of control over daily routines [11]. Chronic exposure to stress can lead to psychological disorders such as depression and is closely linked to increased absenteeism and staff turnover [21]. Notably, the relationship between telework and engagement remains ambiguous: while some workers report higher satisfaction and increased commitment when working remotely [48], others highlight the challenges of isolation and sense of disconnection from their teams [22,27].

In light of this evidence, the implementation of targeted strategies to prevent stress and foster work engagement in remote work settings has become imperative [1]. One of the most effective approaches, according to [64], is the cultivation of an organizational culture that values social support, transparent communication, and trust. Encouraging interaction among employees through digital communication tools that facilitate collaboration and experience sharing is also essential for strengthening team cohesion and promoting engagement in virtual work environments.

#### 3.2.4. Cluster 4—Well-Being, Social Relationships, and Social Support in Remote Work

Cluster 4, in yellow, explores key actions for fostering a positive relationship between remote work and quality of life, with an emphasis on social interactions and the role of support networks [41]. The transition to remote work has introduced concerns that extend beyond productivity to encompass workers’ psychological well-being. Social support in workplace emerges as a crucial element, as it involves the networks formed to buffer the negative effects of work on employees’ health and performance [20,65].

Workplace isolation presents significant barriers to the formation of interpersonal bonds [17] and the development of supportive networks within organizational environments [10]. This isolation is a form of psychological detachment that may persist even when an employee is physically present, yet feels disconnected from daily organizational activities [41]. In remote work settings, interactions are primarily mediated through digital technologies [55], which can reduce the depth and quality of interpersonal relationships [39,54]. When virtual exchanges fall short of fostering a sense of collaboration and belonging, a sense of invisibility may develop, undermining both motivation and willingness to engage with coworkers. Ref. [58] supports this perspective, noting that the absence of physical contact can leave workers feeling unprotected, a condition that may, over time, lead to social withdrawal and reluctance to seek support, either professionally or personally. Ref. [39] further highlights that the lack of face-to-face interaction limits the effectiveness of collaboration and trust-building, thereby constraining problem-solving and creativity. In alignment with this, Ref. [7] observes that insufficient technical and emotional support from organizations contributes to increased stress and burnout, ultimately fostering a perception of social exclusion among remote workers. To counteract this, the author stresses the importance of tailored organizational support that meets workers’ specific needs and expectations.

Workplace well-being initiatives are reflections of broader organizational culture and exert a measurable influence on company success [66]. These practices are closely linked to individual performance, motivation, and engagement [1]. According to [8], several factors directly affect employee well-being, including team relationships [17], effective communication by human resources [58], routine virtual meetings [39], and the presence of programs supporting physical and emotional health. Promoting the physical, emotional, and social well-being of employees is thus not only a strategic priority but also a corporate responsibility that reinforces and generates positive outcomes across all operational dimensions.

The study by [41], conducted among teleworking healthcare professionals at the South London and Maudsley NHS Foundation Trust, found that in the absence of physical and emotional peer support, employees experienced a significant decline in psychological well-being, increased stress levels, and a reduced capacity to cope with work-related demands. These findings are corroborated by [67], who underline that of a lack of perceived social support contributes to elevated emotional strain and deteriorated well-being. Refs. [41,50] further emphasize the critical importance of organizational support in improving both mental health and social cohesion among remote workers. In short, employee well-being has become a critical success factor in contemporary work environments [28,66]. When neglected, it leads to declines in organizational productivity and performance [64]. Thus, promoting well-being in remote work settings is not merely a matter of employee satisfaction, but it is a fundamental component of sustainable organizational success.

#### 3.2.5. Cluster 5—Working Conditions and Professional Isolation

Cluster 5, in purple, focuses on the relationship between working conditions, occupational health, and the implications of professional isolation on employee behavior. Although remote work offers flexibility and autonomy, it often intensifies feelings of isolation, which negatively affect workers’ motivation, organizational commitment, and engagement [17]. This phenomenon is particularly evident in the teleworking context [59], where physical interactions are replaced by impersonal and digital communication [55]. Several studies have explored the challenges that remote work presents to occupational health. Inadequate infrastructure and an imbalanced work–life dynamic in in-home settings have been associated with negative health outcomes, including mental fatigue [29] and sleep disturbances [6]. These conditions suggest that the remote work environment can jeopardize not only employee productivity but also psychological and physiological well-being.

Ref. [12] analyzed the relationship between working conditions and turnover intentions, revealing that poor remote work conditions contribute to a heightened desire among employees to leave their organizations. Ref. [68] reinforced this by demonstrating that professional isolation, combined with feelings of devaluation or a lack of support, fosters job dissatisfaction, which significantly increases the likelihood of employees seeking alternative employment opportunities. A particularly robust line of research in this cluster examines the predictive power of professional isolation and turnover intentions. Meta-analyses, such as that by [48], indicate that professional isolation in telework settings strongly predicts departure intentions (r = 0.47), surpassing other traditional predictors like job satisfaction (r = 0.38) and organizational commitment (r = −0.42).

According to [69], creating a supportive and functional work environment is essential to maximizing the performance of remote employees. However, many workers lack adequate home office conditions, such as ergonomic furniture or a dedicated workspace, which presents serious obstacles to productivity and mental health [70]. The study by [71] further emphasizes the importance of considering home-based environmental factors when evaluating the effects of telework on mental well-being and output. For instance, the presence of pets, such as dogs, can provide emotional support and mitigate feelings of isolation, while access to outdoor spaces can enhance concentration and psychological balance. Ref. [20], supporting the perspective of [69], found that the absence of appropriate working conditions, both technological and emotional, contributed to a sense of disconnection among remote workers. These findings point to a critical insight: for the benefits of teleworking to be sustainable in the long term, organizations must invest in adequate infrastructure, technological support, and a culture of trust and collaboration [39,58].

#### 3.2.6. Cluster 6—Productivity and Sustainability in Teleworking

Cluster 6, in cyan, examines the intricate relationship between knowledge sharing, innovation, productivity, and sustainability, while highlighting the challenges and opportunities posed by teleworking. Research conducted by [12], involving 1270 employees from the banking, finance, and insurance sectors, concluded that physical isolation and a lack of communication among colleagues negatively affect productivity levels. These findings are corroborated by [58], who further argues that team collaboration and problem-solving capacities may be impaired under remote working conditions, ultimately leading to decreased work output and efficiency.

A key obstacle to productivity in telework environments is the limitation of daily interpersonal contact—a vital process for transferring experience and knowledge [6,72]. According to [43], knowledge becomes meaningful only when it is shared among individuals and applied collectively, transforming individual experiences into organizational knowledge. However, teleworking restricts these knowledge flows by reducing opportunities for informal exchanges and spontaneous collaboration, moments that typically foster learning and innovation within physical office spaces [37]. The lack of immediate feedback and the scarcity of constructive, real-time discussions in remote settings often leads to idea stagnation, limiting the generation of new solutions and innovative thinking [43]. Innovation, as emphasized by [37], thrives on spontaneous interaction and the organic sharing of knowledge, elements that are frequently missing in telework environments.

The process of knowledge sharing is essential for both innovation [73] and the continuous development of employees [43]. Nevertheless, a non-physical presence disrupts workplace conversations [17], impeding the transmission of tacit knowledge and lived experience. Furthermore, feelings of loneliness can undermine employee motivation and willingness to engage in knowledge sharing practices [43]. In response to these challenges, researchers such as [54,55] advocate the implementation of strategic approaches in remote environments, specifically structured virtual meetings and collaborative digital platforms that facilitate the exchange of ideas and co-creation. Creating business environments that actively promote collaboration and mutual learning is thus critical to overcoming social and cognitive barriers. As noted by [43], these environments catalyze individual and organizational growth, making knowledge transfer more inclusive and dynamic.

Ref. [58] classifies the impacts of loneliness and isolation in teleworking on productivity into direct and indirect effects. Direct impacts include communication difficulties, reduced collaboration, lower creativity, and limited knowledge sharing, all of which impair task performance. Indirect impacts consist of increased stress, a weakened sense of organizational belonging, and a heightened risk of burnout. These factors negatively affect concentration, decision-making, and overall well-being [60]. Ref. [64] reinforces this view, asserting that employee performance is closely linked to job satisfaction, motivation, and psychological well-being. Finally, Ref. [28] emphasizes the need for organizational strategies that prioritize employee well-being as a foundational element of sustainable productivity.

### 3.3. Main Theories

Theoretical models employed across the analyzed studies were reviewed and systematized, as presented in Table 6. The predominant framework identified is the Job Demands–Resources model [63], which offers a comprehensive lens through which to examine the effects of loneliness and isolation in teleworking. This model provides a theoretical basis for developing interventions aimed at balancing job demands with available resources, thereby enhancing the experience and well-being of remote workers. According to the JD-R model, motivation and well-being improve when workers have sufficient resources, such as support, autonomy, and adequate infrastructure, to cope with job-related demands. This dynamic fosters a more engaging and productive work environment. Furthermore, the model elucidates the interconnections between job demands, available resources, and outcomes such as performance, occupational stress, and job satisfaction, offering practical guidance for organizations seeking to implement strategic and evidence-based people management practices.

## 4. Discussion

The COVID-19 pandemic marked a turning point in the global adoption of teleworking, establishing it as a prevalent work model in many countries [1,42]. Against a backdrop of rapid technological advancement and organizational transformation [2], scholarly interest has grown around the psychosocial dimensions of remote work, particularly the effects of loneliness and isolation on individual well-being and organizational performance. These adverse experiences have increasingly been associated with anxiety, chronic stress [24], depression, and burnout, all of which pose significant risks to both productivity and organizational sustainability [27].

This systematic and bibliometric review aimed to synthesize and map the scientific output on the interplay between loneliness, isolation, and organizational outcomes within the context of teleworking. Guided by PRISMA principles and supported by VOSviewer for network visualization, this review encompassed 65 articles published between 2000 and 2024 and indexed in the Scopus database. Its primary objective was to identify core patterns and thematic clusters that inform the academic exploration of remote work’s psychosocial impacts, particularly those relating to workers’ mental health and organizational productivity.

The findings indicate a pronounced acceleration in scholarly interest post-2020, coinciding with the shift to remote work. Before the pandemic, studies were limited and largely sporadic. The crisis, however, catalyzed a surge in research, peaking in 2022, reflecting heightened awareness of the psychological and structural challenges posed by digital work environments. The increase in scholarly output observed after 2020 aligns with RQ1, suggesting a widespread and timely academic response to the psychosocial effects of teleworking, particularly in the context of the COVID-19 pandemic and its aftermath. This increase not only mirrors the operational transition across industries but also illustrates a deeper recognition of the emotional and cognitive tolls experienced by teleworkers.

From a methodological standpoint, the predominance of quantitative approaches (73.84%) underscores an empirical focus on measuring the tangible outcomes of remote work across diverse sectors such as academia, healthcare, public administration, and ICT. Nonetheless, the underrepresentation of qualitative and mixed-method research points to a limitation in capturing the rich, subjective nuances of teleworking experiences. Additionally, although the review studies come from 40 countries, the concentration of research in North America and Western Europe, particularly in the United States, reveals an uneven global distribution. This suggests a need for expanding the geographic scope of future studies, especially to underrepresented regions in the Global South, where telework dynamics may differ significantly.

The reviewed articles were predominantly published in journals focusing on psychology, public and occupational health, and organizational behavior. The substantial presence of articles in high-ranking journals (Q1 and Q2) confirms the growing academic recognition of this research field.

To answer the second research question, the results showed that, although teleworking is often associated with increased satisfaction and flexibility [41,85], it simultaneously introduces significant psychological stressors. While some scholars [48,64] highlight the productivity benefits linked to work–life balance, others [41,67] emphasize the detrimental effects of prolonged physical and social disconnection. The literature increasingly recognizes that the lack of face-to-face interaction, the erosion of team cohesion, and the challenges of autonomous work management can contribute to decreased motivation and performance [1,10,12,24].

The cluster analysis revealed six thematic cores reflecting converging academic concerns. Clusters 1 to 3 converge on the multifaceted nature of isolation in teleworking, spanning social, organizational, and professional domains. The first cluster, comprising 26.2% of the sample, focused on the mental health impacts of telework, documenting how emotional and social loneliness diminish resilience and weaken employee engagement. These findings reinforce the inadequacy of digital communication in replicating the social dynamics of the physical workplace. The second cluster examined the specific pressures imposed by the COVID-19 pandemic, particularly on high-stress sectors such as healthcare and education. Studies in this group explored the psychological strain resulting from blurred work–life boundaries, increased demands, and technostress. The convergence of domestic and professional roles, alongside heightened productivity expectations, emerged as a key source of burnout and emotional fatigue [86]. Cluster 3 centered on professional stress and work engagement, underscoring how reduced interpersonal interactions correlate with declines in motivation, dedication, and psychological investment in tasks. The prevalence of the Job Demands–Resources (JD-R) model across this cluster reflects its utility in explaining how imbalances between work demands and resources erode engagement and well-being. Cluster four explored social support and well-being, emphasizing the protective role of informal and institutional support structures. Even in digitally connected environments, perceived emotional and organizational detachment can foster feelings of psychological invisibility. The absence of informal feedback and social inclusion undermines both morale and performance. Cluster 5 turned working conditions into moderators of professional isolation. Studies highlighted how suboptimal home office environments, inadequate technology, and a lack of ergonomic resources can amplify dissatisfaction and increase turnover intentions. The lack of recognition and institutional support in these contexts was shown to be a key predictor of employee disengagement and exit behavior. Cluster 6 focused on productivity and knowledge management. Although some studies reported short-term productivity improvements, others warned of long-term risks, including diminished collaboration, reduced innovation, and impaired knowledge transfer. In particular, the loss of spontaneous interactions and informal learning opportunities was seen as detrimental to organizational learning and innovation capacity. Several studies recommended the use of structured digital environments and collaborative tools to mitigate these effects and foster engagement in remote teams.

The co-word analysis corroborated these findings by revealing the centrality of terms such as mental stress, social isolation, COVID-19, and telework, emphasizing their thematic relevance and conceptual interdependence. The interrelations suggest that loneliness and isolation must be understood not as isolated phenomena, but as deeply embedded within broader psychosocial, technological, and organizational systems. Overall, the results of this bibliometric review reveal the increasing complexity and multidimensionality of remote work. While it offers substantial benefits, flexibility, cost savings, and autonomy, it also introduces significant challenges (RQ2). Loneliness and isolation repeatedly emerge as central mediators, impacting not only individual mental health but also team dynamics, organizational innovation, and long-term sustainability. From a theoretical standpoint (RQ3), the prominence of the Job Demands–Resources (JD-R) model, alongside conceptual frameworks centered on well-being and social support, reflects a concerted effort within the academic community to explain and contextualize the psychosocial impacts of teleworking. Building on this foundation, and addressing RQ4, the literature clearly indicates that addressing these challenges requires integrated strategies that consider the psychological, infrastructural, and managerial dimensions of remote work, reinforcing the importance of supportive environments and effective people management practices.

The reviewed literature underscores the importance of adopting a more critical perspective on the long-term sustainability of remote work, as scholarly debate continues regarding the overall benefits of this work model [68]. While the COVID-19 pandemic significantly accelerated its implementation, more recent developments suggest that several organizations worldwide have begun to scale back, or even fully withdraw, their remote work policies [49]. Persistent challenges such as reduced collaboration, the erosion of organizational culture, limitations in knowledge sharing and innovation, difficulties in integrating new employees and managing performance, and structural inequalities related to infrastructure and access to digital tools continue to pose significant barriers [37,87]. These constraints help explain the recent deceleration in the widespread adoption of telework [88]. Accordingly, the literature increasingly reflects a shared understanding that remote work is not a one-size-fits-all solution. Its effectiveness depends heavily on contextual factors, including the sector of activity, the nature of the role, the organization’s level of maturity, and its technological infrastructure [48]. In light of these considerations, hybrid work models have gained considerable traction [87,89]. These models are emerging as the preferred organizational strategy in today’s work environment, as they are seen to better support interpersonal interaction and foster team cohesion [86].

### 4.1. Theoretical and Practical Implications

From a theoretical point of view, this study advances current knowledge on the psychological and organizational consequences of telework, particularly regarding employee mental health. By consolidating key constructs such as loneliness, isolation, and technostress within established psychological and managerial frameworks, it contributes to the theoretical development of organizational behavior and occupational health in digitally mediated work environments. Notably, the prominence of the Job Demands–Resources (JD-R) model across the reviewed literature underscores its continued relevance in explaining how remote work conditions impact individual strain and motivation. Specifically, this framework highlights the importance of providing adequate job resources (e.g., social support, feedback, autonomy) to buffer the negative effects of increased demands linked to spatial and social disconnection in telework settings. In addition, the integration of complementary frameworks, such as Self-Determination Theory and Social Exchange Theory, illuminates the significance of intrinsic motivation, relatedness, and perceived organizational reciprocity in sustaining employee engagement and well-being during remote work. These theoretical perspectives collectively point to a growing interest in multilevel analysis that accounts for the dynamic interaction between individual psychological processes, interpersonal relationships, and organizational structures.

Although hybrid and remote work arrangements are now widely preferred by employees [86], our findings suggest that they may also evoke ambivalence, contributing to decreased motivation, emotional fatigue, and compromised performance. Consequently, this review calls attention to the need for more nuanced investigations into how telework can be optimized through human-centered digital transformation strategies, including supportive communication systems, social connection mechanisms, and accessible mental health interventions [11]. This study also offers several practical implications, summarized in Table 7, for key stakeholders such as HR professionals, team leaders, executives, IT departments, policymakers, and industry associations.

These recommendations aim to enhance the sustainability and inclusivity of remote work practices by focusing on early detection of distress, structured socialization, mental health support, and ergonomically sound digital work environments. For instance, promotion emotional intelligence training for team leaders and ensuring access to psychological support tools are critical steps toward safeguarding employee well-being. In addition, the implementation of weekly meetings between managers and remote employees, addressing not only task performance but also emotional well-being, can play a crucial role in reducing experiences of loneliness and isolation. Such practices also facilitate the early detection of symptoms related to burnout. In line with this, the development of structured virtual programs aimed at integration and socialization—such as mentoring schemes and digital team-building initiatives—is particularly beneficial for onboarding new employees. Furthermore, given the limitations associated with fully remote work, organizations should adopt a nuanced approach when selecting between remote and hybrid work models. This decision should be informed by factors such as the specific sector, the nature and interdependence of the tasks, and the individual characteristics of the workforce. Tailoring the telework arrangement to the complexity of job roles and employee profiles may enhance both the effectiveness and long-term sustainability of hybrid work structures. Finally, it is recommended that organizations adopt a proactive and preventive stance in the implementation of telework, seeking to anticipate potential challenges and tailor interventions to the specific needs and circumstances of remote workers. This includes establishing mechanisms for the continuous monitoring of remote workers’ experiences—such as conducting regular feedback interviews to explore emerging challenges or administering monthly surveys that assess employee well-being, the quality of social interactions, and perceptions of organizational support. By intentionally embedding well-being considerations into the design of digital work environments, companies can enhance not only individual performance and retention but also contribute to the creation of more humane, resilient, and sustainable workplaces in the digital age.

### 4.2. Limitations

Despite the methodological rigor applied in the selection and analysis of relevant literature, this study is subject to several limitations. First, the exclusive reliance on the Scopus database may have resulted in the omission of relevant studies indexed in other databases, such as Web of Science or PubMed. Although Scopus offers broad disciplinary coverage, its indexing criteria may introduce bias toward certain regions or research traditions. Second, the inclusion criteria limited the sample to articles published in English, Portuguese, Spanish, and French, potentially excluding valuable contributions in other languages and constraining the global representativeness of the review. Furthermore, while the review spans the period from 2000 to 2024, most of the relevant literature emerged after 2020, coinciding with the COVID-19 pandemic. This temporal concentration, while reflective of a real-world shift in work modalities, also limits the availability of longitudinal perspectives on remote work. In addition, many of included studies employed cross-sectional designs, which restrict causal inference and fail to capture the evolving nature of psychological and organizational responses to telework. Conceptual and methodological inconsistencies also pose challenges. Key constructs such as loneliness, isolation, and performance are defined and measured in diverse ways across studies, complicating cross-study comparisons and meta-analytical synthesis. Lastly, the literature is disproportionately focused on knowledge-intensive sectors such as ICT, healthcare, and academia, leaving underexplored the realities of telework in other sectors like manufacturing, logistics, and retail.

### 4.3. Future Research Lines

Building on the limitations identified, several promising avenues for future research emerge. First, longitudinal studies are needed to examine how feelings of loneliness and isolation develop over time and how they interact with mental health, job satisfaction, and productivity. Such designs would offer stronger causal insights and support the formulation of sustainable remote work policies. Second, cross-cultural and comparative research is essential to understand how sociocultural norms and institutional factors shape the experience and impact of remote work. Studies that incorporate individual-level moderators, such as age, digital literacy, personality traits, and caregiving responsibilities, can offer valuable guidance for the design of tailored interventions. Another underexplored area involves the systematic evaluation of organizational practices aimed at mitigating isolation and enhancing engagement, such as virtual well-being programs, online rituals, and digital leadership initiatives. Empirical assessments of these interventions are crucial for evidence-based decision-making. Sector-specific studies are also warranted to examine the distinct challenges and opportunities of remote work in industries that remain underrepresented in current research. Additionally, the development and adoption of standardized, validated instruments to measure constructs such as telework-related loneliness, perceived social support, and virtual team dynamics would enhance the comparability, reliability, and cumulative value of future studies. Finally, considering the growing need for a critical reflection on the long-term sustainability of remote work, future research should explore more deeply the underlying factors contributing to the reduction or withdrawal of telework policies across different organizational contexts. Such investigations should aim to uncover the organizational, technological, and cultural drivers shaping this shift. In parallel, there is a pressing need to develop and validate robust instruments capable of consistently measuring the impact of remote and hybrid work models on key dimensions such as organizational culture, employee performance, and overall well-being.

## 5. Conclusions

The COVID-19 pandemic has significantly reshaped the landscape of work, accelerating the adoption of remote and hybrid work models across sectors and regions [1]. Originally introduced as a short-term response to public health emergencies, teleworking has become a lasting organizational arrangement, raising complex questions about its implications for psychological well-being and organizational performance. This review synthesized findings from 65 peer-reviewed articles indexed in Scopus to explore how loneliness and isolation in remote work contexts affect individual and organizational outcomes. The surge of publications post-2020 reflects the urgency with which academia has responded to these emerging challenges. Six thematic clusters were identified, revealing the multidimensional nature of telework’s psychological and operational impacts—from emotional well-being and engagement to knowledge sharing and sustainability. The findings suggest that the absence of face-to-face interaction and informal socialization in remote work settings can result in significant emotional strain, diminished creativity, and reduced performance. While remote work offers benefits in terms of flexibility and autonomy, its psychological risks must not be underestimated. Organizations that neglect these human dimensions may experience reduced employee satisfaction, increased turnover, and long-term reputational costs.

Beyond academic contributions, this review provides actionable insights for organizational leaders, HR professionals, and policymakers. The effective implementation of remote and hybrid work requires more than technological readiness—it demands deliberate strategies to foster social connection, emotional resilience, and inclusive organizational cultures. Policymakers, in turn, must adapt labor standards to recognize isolation and disconnection as occupational health risks in the digital era

In alignment with the United Nations Sustainable Development Goals, particularly SDG 3 (Good Health and Well-being), SDG 8 (Decent Work and Economic Growth), and SDG 10 (Reduced Inequalities), this study advocates organizational practices that promote psychosocial safety, flexibility, and equity in digital work environments. Initiatives such as emotional support programs, peer interaction platforms, and inclusive digital infrastructure are not only feasible but essential to organizational resilience and cohesion in the post-pandemic world.

In conclusion, the future of work will depend not solely on digital tools and platforms, but on the ability of organizations to humanize remote work—by designing systems that protect well-being, promote social connection, and cultivate a shared sense of purpose and belonging, even at a distance.

## Figures and Tables

**Figure 1 healthcare-13-01943-f001:**
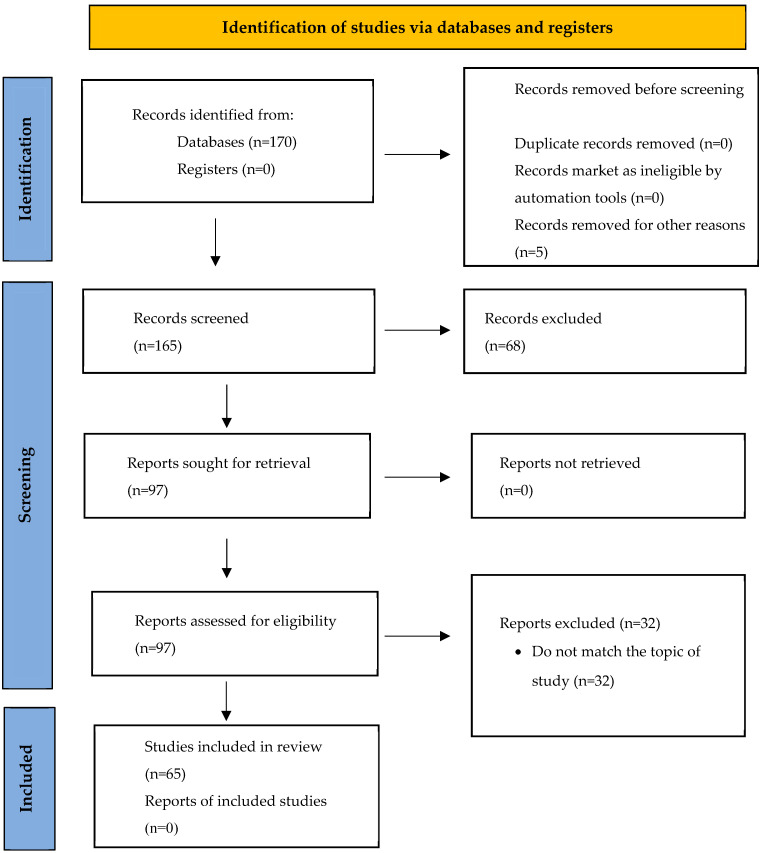
Article selection and exclusion flowchart, according to PRISMA 2020.

**Figure 2 healthcare-13-01943-f002:**
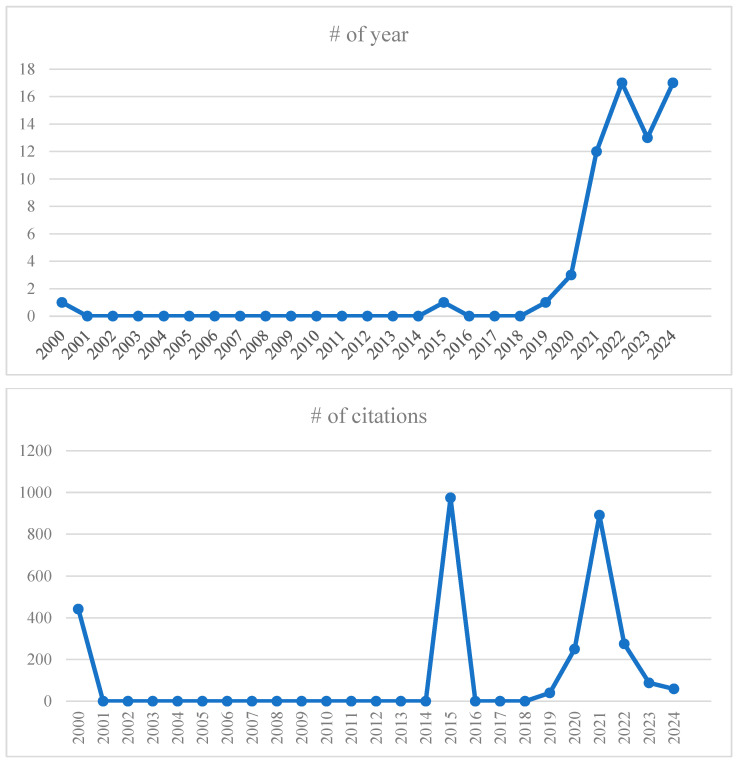
Scientific articles and citations per year.

**Figure 3 healthcare-13-01943-f003:**
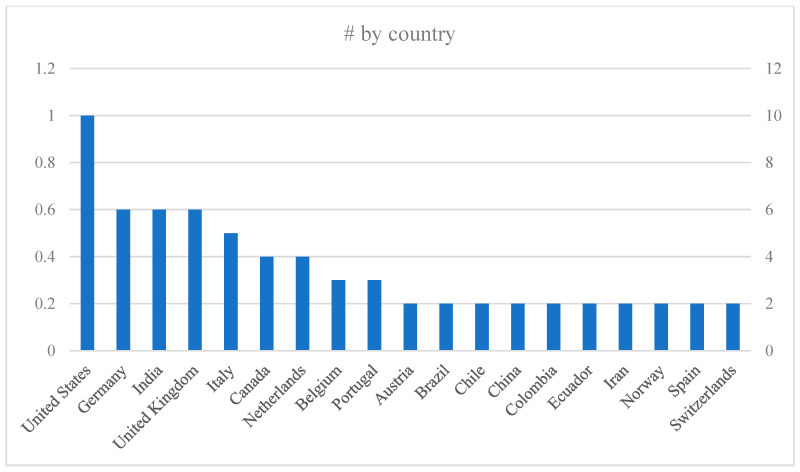
Countries with the highest number of publications.

**Figure 4 healthcare-13-01943-f004:**
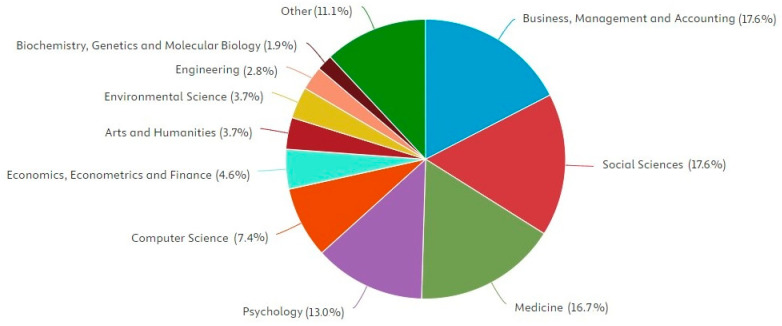
Documents by subject area.

**Figure 5 healthcare-13-01943-f005:**
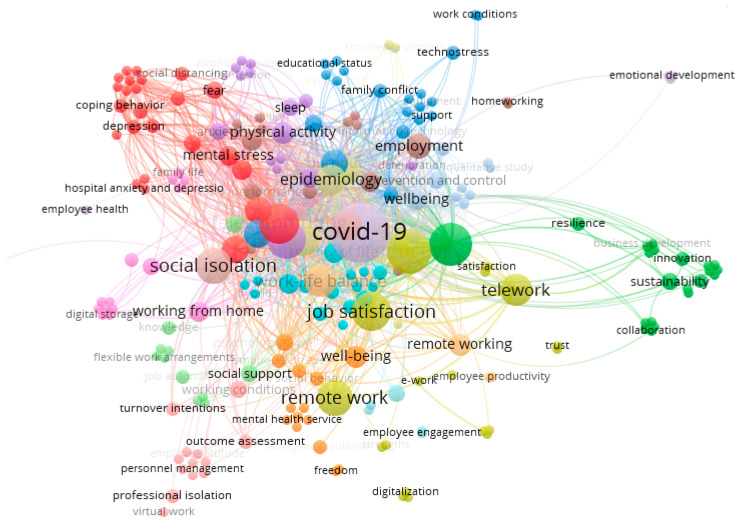
Co-keyword map of isolation and loneliness in relation to telework (threshold: one co-occurrence per keyword).

**Figure 6 healthcare-13-01943-f006:**
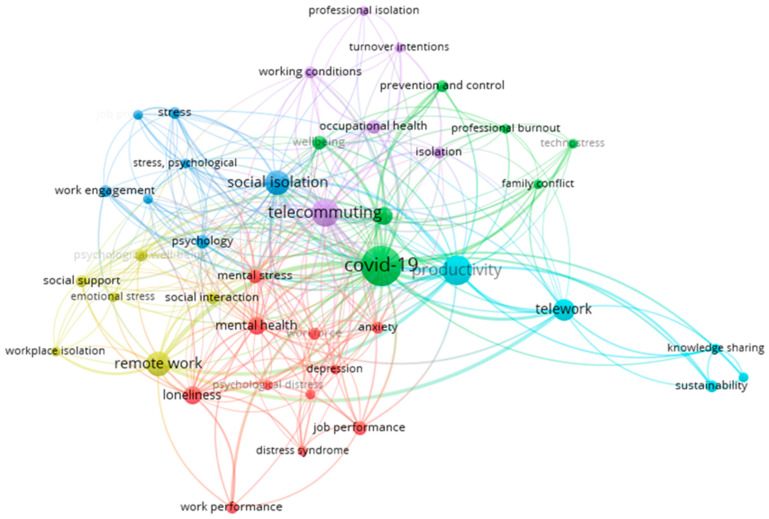
Co-keyword map of isolation and loneliness in relation to telework (threshold: three co-occurrences per keyword).

**Table 1 healthcare-13-01943-t001:** A description of the procedures adopted in the systematic research.

Criteria	Selection	Exclusion
Database: Scopus	TITLE-ABS-KEY (telework* OR “home office” OR “work from home” OR telecommuting OR “virtual work” OR “remote work”) AND TITLE-ABS-KEY (loneliness OR isolation OR solitude OR “social isolation”) AND TITLE-ABS-KEY (performance OR productivity OR “work performance” OR “organizational performance”)	
Document type	Quantitative studies	
	Empirical studies	
	Case studies	Dissertations (Master and Doctorate)
	Qualitative studies with application of interviews or questionnaires	Conferences
	Literature review	Book chapters
	Systematic review	
Language	English, Portuguese, French, and Spanish	
Publication year	From 1 January 2000 to December 2024	
Subject	Telework, loneliness, isolation, performance, productivity	
Results	Motivation	
	Recognition	
	Job satisfaction	
	Performance	
	Engagement	
	Productivity	
	Innovation	
	Creativity	
	Knowledge sharing	
	Feedback	
	Autonomy	
	Self-esteem	
	Safety	
	Reduction in emotional exhaustion	
	Professional valorization	
	Responsibility	
	Skills	
	Quality	
	Loneliness	
	Social isolation	
	Burnout	
	Boreout	
	Frustration	
	Fatigue	
	Workaholism	
	Boredom	
	Monotony	
	Stress	
	Injustice	
	Discouragement	
	Disinterest	

**Table 2 healthcare-13-01943-t002:** Description of type of methodology and sample.

Research Method	%	Sample	# of Articles
Qualitative	12.31	University Workers	2
		ICT Workers	2
		General Workers	4
Quantitative	73.84	Academic Staff	3
		ICT Workers	4
		Public Workers	2
		Industry Workers	3
		Bank Workers	2
		Healthcare Workers	4
		General Workers	26
		Government Workers	3
		Enterprise Workers	1
Mixed-method	1.53	Healthcare Workers	1
Non-empirical	12.31		8

**Table 3 healthcare-13-01943-t003:** The 20 top journals, according to data provided by bibliometric analysis, the Scimago Journal & Country Rank (SJR), and the Journal Citation Report (JCR).

Journal	H-Index SJR	SJR 2023	JIF 2023	Total Citation 2023	Quartile	Research Area	Country	Articles No
*Academic Radiology*	110	1.06	3.8	9023	Q1	Medicine	USA	1
*Applied Psychology*	111	2.66	4.9	6045	Q1	Psychology	UK	1
*Asia-Pacific Journal of Business Administration*	27	0.76	3.3	857	Q1	Business, Manag./Accounting	UK	1
*Behaviour and Information Technology*	95	1.01	2.9	5519	Q1	Psychology/Social science	UK	1
*BJPsych Open*	44	1.46	3.9	3144	Q1	Medicine/Psychiatry	UK	1
*Clinical and Translational Radiation Oncology*	31	1	2.7	1801	Q1	Medicine	Ireland	1
*Cyberpsychology, Behavior, and Social Networking*	180	1.44	N/A	N/A	Q1	Psychology/Social Sciences/Medicine	USA	1
*Empirical Software Engineering*	93	1.51	3.5	5093	Q1	Computer Science	Netherlands	1
*European Journal of Health Economics*	67	1.08	3.1	3700	Q1	Economics/Medicine	Germany	1
*Frontiers in Public Health*	101	0.9	3.0	38,788	Q1	Medicine	Switzerland	1
*Geoforum*	141	1.34	3.4	11,407	Q1	Social Sciences	UK	1
*Global Business and Organizational Excellence*	25	1.15	N/A	N/A	Q1	Business/Manag./Accounting	USA	1
*Human Resource Development International*	61	1.47	3.8	1732	Q1	Business/Manag./Accounting	UK	1
*Human Resource Development Quarterly*	78	1.22	4.0	1963	Q1	Business/Manag./Accounting	USA	1
*IEEE Access*	242	0.96	3.4	244,906	Q1	Computer Science	USA	1
*Information Technology and People*	76	1.24	4.9	3763	Q1	Computer Science/Social Sciences	UK	1
*International Journal of Manpower*	73	1.25	4.6	3311	Q1	Business/Manag./Account.	UK	1
*Journal of Occupational Health Psychology*	146	2.17	5.9	8538	Q1	Psychology/Medicine	USA	1
*Journal of Sleep Research*	139	1.41	3.4	9541	Q1		UK	1
*Kybernetes*	53	0.57	2.5	3515	Q1	Computer Science/Social Sciences	UK	1
*Mayo Clinic Proceedings*	214	1.78	7.2	20,223	Q1	Medicine	UK	1
*Personnel Psychology*	167	3.76	4.7	10,066	Q1	Business/Manag./Account/Psychology	USA	1
*PLoS ONE*	435	0.84	2.9	808,083	Q1	Multidisciplinary	USA	1
*Psychological Science in the Public Interest*	60	9.89	18.2	2770	Q1	Psychology	USA	1
*Sustainability*	169	0.67	3.3	229,272	Q1	Computer Sciences/Social Sciences	Switzerland	1

**Table 4 healthcare-13-01943-t004:** List of authors with more publications and respective metrics.

Total Number of Published Articles	Articles on Topic	Authors	Authors’ Metrics		Affiliation
			H-Index	Citations	
31	2	Toscano, F.	12	1032	Università degli Studi della Campania Luigi Vanvitelli, Italy
57	2	Zappalà, S.	16	1466	Alma Mater Studiorum Università di Bologna, Italy
54	1	Aboelmaged, M.	20	2377	University of Sharjah, United Arab Emirates
196	1	Allen, T.D.	71	21,689	University of South Florida, USA

**Table 5 healthcare-13-01943-t005:** Summary of cluster analysis.

Cluster	Keywords	% Articles	Example of Article	Reference
1. Mental health and job performance	Anxiety, Anguish, Depression, Job performance, Psychological suffering, Loneliness, Mental stress	26.2%	Gore, M.N. (2024). Loss of work–life balance, experience of stress and anxiety among professionals working from home—an exploratory study in a Western Indian city.	[24]
2. COVID-19 and impact on work	COVID-19, Telework, Professional burnout, Family conflict, Prevention and control, Technostress	18.1%	Tobia, L., Vittorini, P., Di Battista, G., D’Onofrio, S., Mastrangeli, G., Di Benedetto, P., & Fabiani L. (2024). Study on psychological stress perceived among employees in an Italian university during mandatory and voluntary remote working during and after the COVID-19 pandemic.	[10]
3. Professional stress and work engagement	Professional stress, Psychology, Social isolation, Motivation, Work engagement	16.4%	Galanti, T., Guidetti, G., Mazzei, E., Zappalà, S., & Toscano, F. (2021). Work from home during the COVID-19 outbreak: the impact on employees’ remote work productivity, engagement, and stress.	[38]
4. Well-being, social relationships, and social support in remote work	Emotional stress, Psychological well-being, Social interaction, Social support, Isolation in the workplace, Telework	14.3%	O’Hare, D., Gaughran, F., Stewart, R., & Pinto da Costa, M. (2024). A cross-sectional investigation on remote working, loneliness, workplace isolation, well-being, and perceived social support in healthcare workers.	[41]
5. Working conditions and professional isolation	Professional isolation, Occupational health, Telework, Working conditions, Turnover intentions	13.8%	Nemțeanu, M.-S., & Dabija, D.-C. (2023). Negative impact of telework, job insecurity, and work–life conflict on employee behavior.	[42]
6. Productivity and sustainability in teleworking	Productivity, Innovation, Knowledge sharing, Sustainability	11.2%	Hodzic, S., Prem, R., Nielson, C., & Kubicek, B. (2024). When telework is a burden rather than a perk. The roles of knowledge sharing and supervisor social support in mitigating adverse effects of telework during the COVID-19 pandemic.	[43]

**Table 6 healthcare-13-01943-t006:** Main theories.

No	Theory	Articles Example
1	Maslow’s Theory of Need (also known as Maslow’s Hierarchy of Needs Theory) [74]—It helps to understand that satisfying human needs is not limited to work itself, but also to the social interactions and emotional support that the workplace provides. Thus, this theory helps to identify the areas that need intervention to reduce the impact of loneliness and isolation in teleworking.	[55]
5	Self-Determination Theory [75]—This theory focuses on 3 basic psychological needs, autonomy, competence, and relatedness, and allows us to explore how the workplace affects psychological needs and how these needs can, in turn, influence mental health, motivation, and performance in remote work.	[11,37,42,43,50]
5	Social Exchange Theory [76]—This suggests that social interactions in the workplace, such as feedback, social support, and recognition, influence workers’ well-being.	[20,37,41,42,50]
1	Agency Theory [77,78]—This analyzes the relationship between the principal (person who delegates the functions) and the agent (person who executes them), focusing on conflicts of interest that may arise due to the difference in objectives between both parties. This theory allows us to understand how the dynamics of supervision and communication in teleworking can affect a worker’s behavior.	[3]
10	Job Demands–Resources Model [63]—This highlights the need for a balance between work requests and available resources to improve the worker experience.	[12,36,38,48,49,52,59,60,64,68]
1	Compensatory Internet Use Theory [79]—This suggests that individuals use the internet as a resource that helps to meet emotional and social needs.	[80]
1	Cognitive Load Theory [81]—This argues that the mind has a limited capacity, and therefore, the amount of information to be processed must be optimized to facilitate learning.	[36]
2	Conservation of Resources Theory [82]—Individuals seek to protect and conserve their resources, aiming for well-being. According to this theory, stress arises when there is a threat or loss of these resources.	[42,48]
1	Leader–Member Exchange [83]—This explores the quality of the relationship between leader and “led”, suggesting that high-quality relationships can lead to better results at work.	[84]

**Table 7 healthcare-13-01943-t007:** Target audience recommendation purpose.

Target	Recommendation	Goal
HR Managers	Implement structured virtual socialization (e.g., online coffee breaks, informal chat groups)	To reduce feelings of isolation and build team cohesion
Team Leaders	Provide training in emotional intelligence and remote team management/leadership	To identify early signs of distress and maintain engagement
Organizations (general)	Define and enforce clear boundaries between work and personal time	To minimize burnout and preserve work–life balance
Executives	Invest in digital well-being tools and mental health support (e.g., EAPs, access to therapists)	To promote psychological safety and resilience
IT & Facilities Teams	Assess and support ergonomic and technological conditions of employees’ home workspaces	To ensure equitable and productive working conditions
Policymakers	Update labor laws to address remote work-related risks (e.g., isolation, technostress)	To establish minimum standards for remote work environments and mental health protections
Industry Associations	Promote sector-specific best practices and guidelines for sustainable telework	To ensure competitive advantage while safeguarding employee well-being

## Data Availability

The data supporting the reported results can be made available upon request to the authors.

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
