# Peer review of "Loneliness and Isolation in the Era of Telework: A Comprehensive Review of Challenges for Organizational Success"

_healthcare, 2025, doi:10.3390/healthcare13161943_

Round 1

Reviewer 1 Report

Comments and Suggestions for Authors

The manuscript is really well structured and written. An initial review of the work indicates the author's dedicated approach and keen sense of the details that make a difference. The manuscript is characterized by a high academic level and essentially no weak points. The introduction is well designed with a clear definition of the research objective and research questions. The Methodology section is detailed, convincing and clear, with good graphics. The authors presented the limitations of the used methodology in a fair way in the Limitations section at the end of the paper. Despite all these good aspects of the manuscript there is room for some improvements:

1. In the "Discussion" section, it is necessary to address the research questions defined in the Introduction in a more explicit way.

2. Also in the "Discussion" section, it is necessary to take a critical attitude towards remote work. After the Covid-19 pandemic, many companies around the world have rapidly shown a tendency to eliminate the option of remote work. The discussion should be critically oriented and it should respect the "pros" and "cons" arguments for remote working.
In accordance with the critical review, the "Future research lines" section should also be revised.

3. The practical implications are too generic, without real proposals and managerial implications. The impression is that this section needs a more extra attention.

In general, the authors have done an excellent job and all they have to do is fine-tune the manuscript. The writing style is excellent, as is the English. Although the work is not based on empirical research, it is possible to recognize its potential for readers.

Author Response

Comment: In the "Discussion" section, it is necessary to address the research questions defined in the Introduction in a more explicit way.

Response: Thank you for pointing this out. We have accordingly revised this issue. Please see in red.

Comment: Also in the "Discussion" section, it is necessary to take a critical attitude towards remote work. After the Covid-19 pandemic, many companies around the world have rapidly shown a tendency to eliminate the option of remote work. The discussion should be critically oriented and it should respect the "pros" and "cons" arguments for remote working.

Response: Thank you for pointing this out. The reviewer is absolutely right, and we have acted accordingly. Please see in red.

Comment: In accordance with the critical review, the "Future research lines" section should also be revised.

Response: Thank you for this suggestion. This section has been duly revised.

Comment: The practical implications are too generic, without real proposals and managerial implications. The impression is that this section needs a more extra attention.

Response: Thank you for pointing this out. We delved deeper into the implications, which made the paper much more robust. Please, see in red.

Reviewer 2 Report

Comments and Suggestions for Authors

See attached report.

Author Response

Comment: Lines 34-35: Please clarify whether the terms “mental” and “psychological” are used synonymously or if you intend to distinguish between them – and if so, how.

Response: Thank you for pointing this out. This issue has been resolved and the mention of "psychological" health has been removed.

Comment: The Methods section partially includes results. I recommend moving all findings to a new section titled “Results”.

Response: Thank you for this suggestion. We have acted accordingly.

Comment: A clear operational definition of the main variables (“telework”, “loneliness”, “isolation”) is lacking. Please, define each term and explain how these definitions shaped your eligibility criteria.

Response: Thank you for this valuable suggestion. We have acted accordingly. Please see in red.

Comment: Inclusion and exclusion criteria are vaguely reported. I suggest providing a detailed table or appendix listing all excluded studies with reasons for exclusion, as per PRISMA standards.

Response: Thank you for pointing this out. We have accordingly revised this section. However, we believe that since the inclusion and exclusion criteria are clearly defined, a table or appendix is not necessary.

Comment: The Boolean search string is described, but the rationale for specific criteria (languages, publication years) is not explained. I suggest justifying why only English, Portuguese, French, and Spanish were included, and why the timeframe was set from 2000-2024.

Response: Thank you for pointing this out. We present a justification for the choice of languages. It is worth noting that these four languages are different, which reduces the possible source of bias associated with including only one. A justification for the selected time period was also provided. (see in red).

Comment: It is unclear how disagreements between reviewers were resolved and whether inter-rater reliability was assessed. Please, report how conflicts were managed and include any kappa statistics or agreement rates if available.

Response: Thank you for pointing this out. The appropriate justifications have been added. Please, see in red.

Comment: Several variables discussed in the findings (e.g., citation counts, countries, journals, theoretical frameworks) were not specified in the data extraction section. I recommend that you clearly list all variables extracted and their relevance to the research questions.

Response: Thank you for this valuable suggestion. We have acted accordingly. Please, see in red.

Comment: Clarity is needed on how the thematic cluster analysis was performed. Please, describe how clusters were formed.

Response: Thank you for pointing this out. The description has been duly added. Please, see in red. It should be noted that VOSviewer generates clusters by visualizing similarities between items (e.g., keywords) in a network, and then grouping them based on their proximity and connections. The software uses layout algorithms to position items on a map, and then clustering techniques to identify groups of items that are closely related.

Comment: Clarify the inclusion of students in the “worker” sample. Does this align with your definition of “teleworkers”?

Response: Thank you for pointing this out. This issue has been accordingly revised.

Round 2

Reviewer 2 Report

Comments and Suggestions for Authors

Dear Authors, thanks for your willingness to consider and apply my feedback. Next time, please indicate explicitly the lines where you have made modifications. Good luck with your work!

Author Response

Comment: Dear Authors, thanks for your willingness to consider and apply my feedback. Next time, please indicate explicitly the lines where you have made modifications. Good luck with your work!

Response: Thank you for your insightful comments and suggestions. The article has become academically more robust. In the future, we will take your recommendation into account. Thank you.